# DISTANTLY SUPERVISED RELATION EXTRACTION IN FEDERATED SETTINGS

## ABSTRACT

Distant supervision is widely used in relation extraction in order to create a large-scale training dataset by aligning a knowledge base with unstructured text. Most existing studies in this field have assumed there is a great deal of centralized unstructured text. However, in practice, text may be distributed on different platforms and cannot be centralized due to privacy restrictions. Therefore, it is worthwhile to investigate distant supervision in the federated learning paradigm, which decouples the training of the model from the need for direct access to the raw text. However, overcoming label noise of distant supervision becomes more difficult in federated settings, because the sentences containing the same entity pair scatter around different platforms. In this paper, we propose a federated denoising framework to suppress label noise in federated settings. The core of this framework is a multiple instance learning based denoising method that is able to select reliable sentences via cross-platform collaboration. Various experimental results on New York Times dataset and miRNA gene regulation relation dataset demonstrate the effectiveness of the proposed method.

## 1 INTRODUCTION

Relation extraction (RE) aims to mine factual knowledge from free text by labeling relations between entity mentions, which is a crucial step in knowledge base (KB) construction. For example, given a sentence "$[Steve\ Jobs]_{e_1}$ and Wozniak co-founded $[Apple]_{e_2}$ in 1967", a relation extractor should identify that "*Steve Jobs*" and "*Apple*" are in a "*Founder*" relationship.

Most existing supervised RE systems, such as Zeng et al. (2014); Zhang & Wang (2015); Wang et al. (2016); Zhou et al. (2016), rely on a large-scale manually annotated training dataset, which is extremely expensive and cannot cover all walks of life. To ease the reliance on annotated data, Mintz et al. (2009) proposed distant supervision to automatically generate training data by heuristically aligning a KB with unstructured text. The key assumption of distant supervision is that if two entities have a relation in the KB, then all sentences that mention these two entities will express this relation. Since then, there has been a rich literature devoted to this topic, such as Riedel et al. (2010); Hoffmann et al. (2011); Zeng et al. (2015); Lin et al. (2016); Ye & Ling (2019); Yuan et al. (2019).

Though the progress is exciting, distant supervision approaches have so far been limited to the centralized learning paradigm, which assumes that a great deal of text is easily accessible. However, in practice, text may be distributed on different platforms and be massively convoluted with sensitive personal information, especially in the healthcare and financial fields (Yang et al., 2019; Zerka et al., 2020; Chamikara et al., 2020). Due to privacy restrictions, it is almost impossible or cost-prohibitive to centralize text from multiple platforms. Recently, federated learning (McMahan et al., 2016) provides a compelling solution for learning a model from decentralized and privacy-sensitive data. The main idea behind federated learning is that each platform trains a local model based on its own local data and a master server coordinates massive platforms to collaboratively train a global model by aggregating these local model updates.

Unfortunately, directly applying federated learning to the decentralized distantly supervised data fails, because conventional federated learning requires the local data to come with labels without noise (Tuor et al., 2020), however, in distant supervision, automatic labeling inevitably accompanies with **label noise** (Riedel et al., 2010; Hoffmann et al., 2011; Zeng et al., 2015; Lin et al., 2016),

which means not all sentences that mention an entity pair can represent the relation between them. Training on such noisy data will substantially hinder the performance of the RE model.

Moreover, even involving previous denoising methods, such as Zeng et al. (2015); Lin et al. (2016); Ye & Ling (2019), cannot handle label noise well in federated settings. This point can be illustrated by the example in Figure 1. $S_1$ and $S_2$ contain the same entity pair (*"Steve Jobs"*, *"Apple"*) but are distributed on two platforms. $S_1$ is true positive while $S_2$ is a false positive instance, which does not express the *"founder"* relation. In centralized training, there is no barrier between Platform 1 and Platform 2; therefore, simultaneously considering $S_1$ and $S_2$ can easily filter out noise via only selecting $S_1$ (Zeng et al., 2015) or placing a small weight on $S_2$ (Lin et al., 2016; Ye & Ling, 2019). However, raw data exchange between

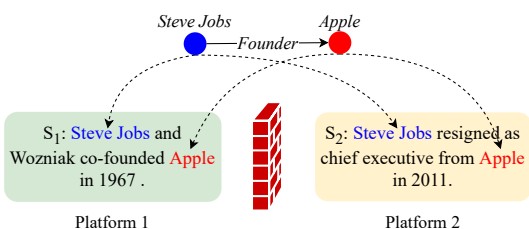

Figure 1: An example of the sentences that contain the same entity pair distributed on two platforms. The triple (*Steve Jobs*, *Founder*, *Apple*) is a fact in the KB

platforms is prohibited in federated settings. Due to the lack of comparison with $S_1$, previous denoising methods would mistakenly regard $S_2$ as a true positive instance. As a result, $S_2$ is retained and then poisons the local model in platform 2, which would affect the global model in turn.

To suppress label noise in federated settings, we propose a federated denoising framework in this paper. The core of this framework is a multiple instance learning (MIL) (Dietterich et al., 1997; Maron & Lozano-Pérez, 1998) based denoising algorithm, called **Lazy MIL**, which is only executed at the beginning of each communication round and then would rest until the next round. Since the sentences containing the same entity pair scatter around different platforms, Lazy MIL algorithm coordinates multiple platforms to jointly select reliable sentences. Once sentences have been selected, they would be used repeatedly to train local models until the end of this round.

In summary, the contributions of this paper are:

- Considering data decentralization and privacy protection, we investigate distant supervision under the federated learning paradigm, which decouples the model training from the need for direct access to the raw data. To our best knowledge, combining federated learning with distant supervision is still an unexplored territory, which is the main focus of this paper.

- Since the automatic labeling in distant supervision inevitably accompanies with label noise, we present a multiple instance learning based denoising method, which can select reliable instances via cross-platform collaboration.

- The proposed method yields promising results on two benchmarks datasets, and we perform various experiments to verify the effectiveness of the proposed method. The code will be released at http://anonymized.

## 2 RELATED WORK

In this section, we will briefly review the recent progress in distant supervision and some existing studies in federated learning.

**Distant supervision**. Relation extraction is a task of mining factual knowledge from free text by labeling relations between entity mentions. To alleviate the dependence of supervised methods on annotated data, Mintz et al. (2009) proposed distant supervision by using a knowledge base to annotate a large-scale dataset automatically. However, automatic labeling inevitably accompanies with label noise. To deal with label noise, most distantly supervised approaches (Riedel et al., 2010; Hoffmann et al., 2011; Surdeanu et al., 2012; Zeng et al., 2015; Lin et al., 2016; Luo et al., 2017; Ye & Ling, 2019; Yuan et al., 2019) focus on reducing label noise at bag [1] level prediction. These studies fall under multiple instance learning framework, which assumes that at least one sentence expresses the relation in a bag. Another line of work aims to reduce label noise at sentence level prediction.

---

[1]A set of sentences containing the same entity pair is called a "bag"

These studies (Zeng et al., 2018; Feng et al., 2018; Qin et al., 2018a;b) use reinforcement learning or adversarial training to select trustable relation labels by matching the predicted labels with distantly supervised labels. Compared with previous studies, our work focuses on reducing label noise at bag level prediction and extends distant supervision to federated settings.

Distant supervision has also been applied to other natural language processing tasks, such as named entity recognition (Ghaddar & Langlais, 2018; Shang et al., 2018; Nooralahzadeh et al., 2019), event extraction (Chen et al., 2017), sentiment classification (Go et al., 2009) and question answer (Joshi et al., 2017; Lin et al., 2018; Cheng et al., 2020).

**Federated Learning**. Recently, federated learning (McMahan et al., 2016; Konečnỳ et al., 2016a;b; Bonawitz et al., 2017; Smith et al., 2017; Caldas et al., 2018; Zhao et al., 2018; Li et al., 2018; Jeong et al., 2018; Peng et al., 2019; Li et al., 2019; Wang et al., 2020; Rothchild et al., 2020; Yu et al., 2020) has become a rapidly developing topic in the research community, since it provides a new communication-efficient way of learning a model over a collection of highly distributed platforms while still preserving data privacy. However, most of the previous studies require the data stored by the local platforms to come with ground-truth labels without noise. The problem of how to adapt federated learning to a noisy environment is relatively ignored. In terms of overcoming noise in federated settings, Tuor et al. (2020) is most relevant to our work but require a clean benchmark dataset to train a benchmark model. Compared with Tuor et al. (2020), our work does not rely on a clean benchmark dataset, which does not exist in distant supervision.

# 3 FEDERATED DENOISING FRAMEWORK

## 3.1 TASK DEFINITION

In this paper, we focus on distant supervision in federated settings. Define $K$ platforms $\{P_1, ...P_K\}$ with respective unlabeled corpora $\{D_1, ...D_K\}$. Under the assumption of centralized training, each platform transfers or shares its local corpus to a server, and the server will take the integrated corpus $D = D_1 \cup ... \cup D_K$ to conduct training, while the task of distant supervision in federated settings requires platform $P_i$ does not expose its corpus $D_i$ to others (including the server). In distant supervision, a KB is required to automatically label these corpora. In this paper, we only focus on the data security of these unlabeled corpora and assume the KB is publicly available for all platforms. The issue of protecting the security of KB is beyond the scope of the current work.

To solve this task, we propose a federated denoising framework. The key components of this framework will be elaborated in the following section. Concretely, we firstly introduce the basic relation extractor in Section 3.2, which is the network architecture shared by the global model and local models. Then, we present how to select reliable instances via cross-platform collaboration in Section 3.3. Next, we describe how to use the selected instances to train the local model in Section 3.4. Finally, we present how to use the FedAvg algorithm to update the global model in Section 3.5.

## 3.2 RELATION EXTRACTOR

Following previous studies (Zeng et al., 2015), we adopt the Piecewise Convolutional Neural Network (PCNN) as our relation extractor. Given a sentence $s$ and two entities within this sentence, we first split the sentence into tokens, and then each token $w_i$ is mapped into a dense word embedding $\mathbf{e}_i \in \mathbb{R}^{d_w}$. To specify the entity pair, relative distances between the current token $w_i$ and the two entities are transformed into two positional features by looking up the position embedding matrices. Next, the token is represented as the concatenation of the word embedding and two positional features, and is fed into a convolutional neural network. Then, piecewise max pooling (Zeng et al., 2015) is employed to extract the high-level sentence representation. In the piecewise max pooling, an input sentence is divided into three segments based on the two entities, and the maximum value of CNN outputs in each segment is returned. After that, we apply a single fully connected layer to output the logit value $\mathbf{o}$. Finally, the conditional probability of $j$-th relation is denoted as follows:

$$p(rel_j|s,\Theta) = \frac{exp(\mathbf{o}_j)}{\sum\limits_{i=1}^{M} exp(\mathbf{o}_i)} \tag{1}$$

where $\Theta$ is the model parameter and M is the total number of relation.

---

**Algorithm 1** Lazy Multiple Instance Learning

---
1: **Input**: global model parameters $\Theta$, the set of activated platforms $A$.
2: Define two dictionary on the server, named $V$ and $I$           ▷ Run on the master server
3: Distribute $\Theta$ to each platform in $A$
4: **for** each platform $i \in A$ **in parallel do**           ▷ Run on the activated platforms
5:      **for** each triple $(h, r, t)$ in KB **do**
6:          **for** each sentence $s_z^i$ in the bag $b^i$ **do**
7:              Compute $p(r|s_z^i, \Theta)$           ▷ According to Equation 1
8:          $v^i, id^i \leftarrow \max_z(p(r|s_z^i, \Theta)), s_z^i \in b^i$          ▷ $v^i$ is called uploaded value
9:          Upload $[v^i, id^i, i]$ to the server and append $[v^i, id^i, i]$ to $V[(h, r, t)]$
10: **for** each key $(h, r, t)$ in $V$ **do**           ▷ Run on the master server
11:      Sort $V[(h, r, t)]$ in descending order according to the uploaded values.
12:      $I[(h, r, t)] \leftarrow V[(h, r, t)][0]$
13: Broadcast $I$ to each platform in $A$

---

### 3.3 Lazy Multiple Instance Learning

To avoid the local relation extractor being poisoned by false positive instances, we propose lazy multiple instance learning (Lazy MIL), which can select reliable instances via cross-platform collaboration. The overview of Lazy MIL is illustrated in Algorithm 1.

Suppose that there is a triple $(h, r, t)$ in the public KB, the set of sentences containing the head entity $h$ and tail entity $t$ is represented as $\{(s_1^1, s_2^1, ..., s_{n_1}^1), ..., (s_1^K, s_2^K, ..., s_{n_k}^K)\}$, where $s_i^j$ indicates the $i$-th instance in the platform $j$. In the $q$-th communication round, assume that only platform $i$ and platform $j$ are activated. At the beginning of this round, the parameters of the global model $\Theta_q$ are distributed to the activated platforms $i$ and $j$ for initializing local models, which ensures that all activated local models share the same parameters in Lazy MIL. In platform $i$, the sentences in the set $(s_1^i, s_2^i, ..., s_{n_i}^i)$ are fed into the local model to get conditional probabilities associated with the relation $r$ according to Equation 1, where $r$ is the predicate of the triple. The value $v^i$ and index $id^i$ of the instance with the maximum conditional probability associated with the relation $r$ are computed as follows:

$$v^i, id^i = \max_z(p(r|s_z^i, \Theta_q)) \quad 1 \le z \le n_i \tag{2}$$

After computation, platform $i$ uploads the value $v^i$ and index $id^i$ to the master server. At the same time, the same procedure is performed on platform $j$, and the value $v^j$ and index $id^j$ are also uploaded to the server.

The master server decides which local instance can be selected among all activated platforms based on the uploaded values. If $v^i > v^j$, then the $id^i$-th sentence in platform $i$ is selected as the reliable sentence that expresses the triple $(h, r, t)$ in this round. This decision, called denoising information, is broadcast to all activated platforms. Each activated platform selects reliable training instances from its local corpus according to this denoising information. Note that since only values and indices of conditional probabilities are uploaded to the master server, Lazy MIL almost does not leak the corpus information in each platform.

### 3.4 Local Model Training

After platform $i$ selects reliable instances from its local corpus $D_i$, the selected reliable instance set $D_i^\star$ is used for training the local relation extractor. We use the cross-entropy loss function to optimize parameters $\Theta_q$, which is defined as follows:

$$J(\Theta_q; D_i^\star) = -\frac{1}{|D_i^\star|} \sum_{u=1}^{|D_i^\star|} \log p(r_u|s_u^\star, \Theta_q) \tag{3}$$

where $s_u^\star$ indicates the $u$-th sentence in the selected reliable instance set $D_i^\star$. After training $E$ epochs on the selected reliable instance set, the trained parameters $\Theta_{q+1}^i$ are uploaded to the master server, where the superscript $i$ indicates the parameters are trained on platform $i$.

---

**Algorithm 2** Federated Denoising Framework

---

1: **Hyperparameters**: $K$ is the total number of platforms; $C$ is the fraction of platforms; $B$ is the local minibatch size; $E$ is the number local epochs; $\eta$ is the learning rate.
2: **Master server executes:**
3:     Initialize $\Theta_0$
4:     **for** communication round $q = 0,1,...$ **do**
5:         $m \leftarrow \max(C \times K, 1)$                                     ▷ Select activated platforms
6:         $A_q \leftarrow$ (random set of $m$ platforms)
7:         Execute lazy multiple instance learning algorithm           ▷ Defined in Algorithm 1
8:         **for** each platform $i \in A_q$ **in parallel do**
9:             $\Theta_{q+1}^k \leftarrow$ Local_Training$(i, \Theta_q)$
10:        $\Theta_{q+1} \leftarrow \sum_{i \in A_q} \frac{|D_i^\star|}{\sum_{j \in A_q} |D_j^\star|} \Theta_{q+1}^i$                       ▷ Defined in Equation 5

11:
12: **Function** Local_Training$(i, \Theta)$:                                     ▷ Run on platform $i$
13:     Generate denoised dataset $D_i^\star$ from $D_i$ based on the denoising information $I$
14:     $\mathcal{B} \leftarrow$ (split $D_i^\star$ into batches of size $B$)
15:     **for** each local epoch $e$ from 1 to $E$ **do**
16:         **for** batch $b \in \mathcal{B}$ **do**
17:             $\Theta \leftarrow \Theta - \eta \nabla J(\Theta; b)$                             ▷ $J$ is defined in Equation 3
18:     **return** $\Theta$ to the master server

---

## 3.5 GLOBAL MODEL UPDATE

Suppose $A_q$ is the set of activated platforms in the $q$-th communication round. After all activated platforms finish local training, the master server collects all trained parameters $\{\Theta_{q+1}^i | i \in A_q\}$ to update the global model. We define the goal of the global model as follows:

$$\min_{\Theta_q} \sum_{i \in A_q} \frac{|D_i^\star|}{\sum\limits_{j \in A_q} |D_j^\star|} J(\Theta_q; D_i^\star) \tag{4}$$

where $J(\Theta_q; D_i^\star)$ is the local loss function for the platform $i$. Follow previous studies (McMahan et al., 2016), we optimize this global objective function via taking the weighted average of all trained parameters, which is shown as follows:

$$\Theta_{q+1} = \sum_{i \in A_q} \frac{|D_i^\star|}{\sum\limits_{j \in A_q} |D_j^\star|} \Theta_{q+1}^i \tag{5}$$

where $\Theta_{q+1}^i$ is the optimal parameters obtained by minimizing the local loss function on the local data of platform $i$. Since all trained parameters from different platforms are aggregated together, the corpus information of each platform is hard to be inferred. Thus, corpora in platforms are well-protected. Complete pseudo-code of this framework is given in Algorithm 2.

## 4 EXPERIMENTS

### 4.1 DATASETS AND EVALUATION METRICS

Since conducting experiments on non-public privacy-sensitive datasets is not reproducible, we choose public distantly supervised relation extraction datasets to investigate the effectiveness of the proposed framework.

**NYT 10**[2] (Riedel et al., 2010) is a widely used dataset in distant supervision. It was automatically generated by aligning the semantic triples in Freebase with the New York Times corpus. The training set contains 466,876 sentences, 251,928 entity pairs and 16,444 relational facts. The development set contains 55167 sentences, 28077 entity pairs and 1,808 relational facts. The test set contains

---

[2]https://github.com/thunlp/OpenNRE

172,448 sentences, 96,678 entity pairs and 1,950 relational facts. There are 52 actual relations and a special relation NA for representing no relation between two entities.

**MIRGENE**[3] (Li et al., 2017) is a large-scale biomedical dataset. This dataset is generated by aligning Tarbase and miRTarBase with the abstracts in Medline. An example is shown in the following: " ***MicroRNA-223*** *regulates* ***FOXO1*** *expression and cell proliferation*", where MicroRNA-223 is a miRNA and FOXO1 is a gene. There are 172727 sentences in the training set and 1239 sentences in the test set. We randomly sampled 10% of the data from the training set as the development set.

**Data Partitioning**. To study distant supervision in federated settings, we need to specify how to distribute the data across platforms. In this paper, we focus on the IID situation, where the training data are shuffled and then partitioned into $K$ (the total number of platforms) platforms.

**Evaluation Metrics**. We evaluate our approach and baseline methods on the held-out test set of these two datasets. Precision-recall (PR) curves, area under curve (AUC) values and Precision@N (P@N) values are adopted as evaluation metrics in our experiments.

## 4.2 EXPERIMENTAL SETTINGS

For a fair comparison, we implement our method and all baselines in the same experimental settings. We divide the hyperparameters into three parts, i.e., fixed hyperparameters, unfixed hyperparameters and federated hyperparameters. Fixed hyperparameters follow the hyperparameter settings in Lin et al. (2016), including the 50-dimensional pretrained word embeddings for NYT, the 5-dimensional position embeddings, and CNN module that includes 230 filters with a window size of 3. For MIRGENE, 200-dimensional word embed-

| Hyperparameter | Search Space |
|---|---|
| Learning Rate ($\eta$) | 0.05, 0.08, 0.1,0.2 |
| Learning Rate Decay | 0.01, 0.05 |
| Dropout | 0.1, 0.2, 0.5 |
| Weight Decay | 1e-5, 1e-6 |

Table 1: The search space of unfixed hyperparameter.

dings pretrained on PubMed and MIMIC-III are used. The optimal unfixed hyperparameters are determined by grid search based on the performance of the development set, and the search space of unfixed hyperparameters is shown in Table 1. Federated hyperparameters include the total number of platforms $K$, the fraction of platforms $C$, the local minibatch size $B$, the number of local epochs $E$. All of these control the amount of computation. In the end-to-end comparison, we fix the $K$ to 100, $B$ to 32, $E$ to 3, and set the hyperparameter space of $C$ as $\{0.1, 0.2, 0.5, 1\}$ following McMahan et al. (2016). We use stochastic gradient descent as the local training optimizer and all experiments can be done by using a single GeForce GTX 1080 Ti.

## 4.3 BASELINES

We compare our method with the following baselines in federated settings: (1) Directly applying FedAvg algorithm (McMahan et al., 2016) to the automatically labeled data is the first baseline, which is called **NONE**. In this case, there is no denoising module in this method. (2) Zeng et al. (2015) proposed to leverage multiple instance learning to choose the most reliable sentence as the bag representation, and we abbreviate this method as **ONE**; (3) **ATT** was proposed by Lin et al. (2016), which uses the attention mechanism to select reliable instances by placing soft weights on a set of noisy sentences; (4) **AVE** (Lin et al., 2016) is a naive version of ATT and represents each sentence set as the average vector of sentences inside the set; (5) **ATT_RA** (Ye & Ling, 2019) is a variant of ATT, which calculates the bag representations in a relation-aware way. The federated framework of these baselines is shown in Algorithm 3 in the appendix.

## 4.4 MAIN RESULTS

Figure 2 and Figure 3 show the precision-recall curves on NYT dataset and MIRGENE datasets. At the appendix, we also present AUC values of these curves in Table 8 and detailed precision values measured at different points along these curves in Table 9. To reduce randomness, we run 10

---

[3]https://github.com/leebird/bionlp17

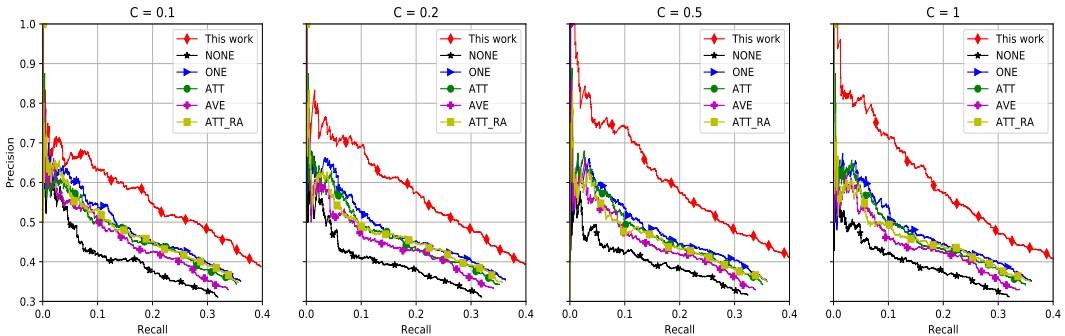

Figure 2: Aggregate precision-recall curves on NYT 10 dataset, where $C$ is the fraction of platforms that are activated on each round.

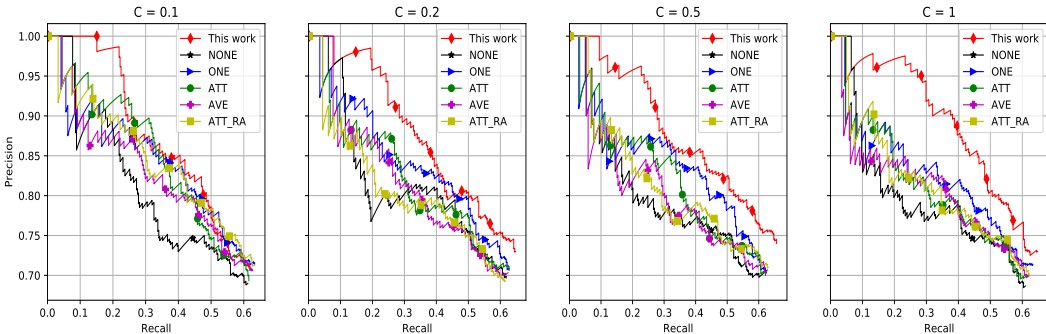

Figure 3: Aggregate precision-recall curves on MIRGENE dataset, where $C$ is the fraction of platforms that are activated on each round.

| AUC | NONE | ONE | ATT | AVE | ATT_RA | Ours |
|---|---|---|---|---|---|---|
| C=0.1 | 0.1287±0.0034 | 0.1719±0.0030 | 0.1638±0.0030 | 0.1521±0.0029 | 0.1664 ±0.0026 | **0.2189±0.0025** |
| C=0.2 | 0.1255±0.0032 | 0.1710±0.0029 | 0.1630±0.0028 | 0.1517±0.0027 | 0.1642±0.0022 | **0.2285±0.0023** |
| C=0.5 | 0.1239±0.0045 | 0.1701±0.0020 | 0.1619±0.0025 | 0.1513±0.0024 | 0.1630±0.0020 | **0.2420±0.0021** |
| C=1.0 | 0.1223±0.0037 | 0.1689±0.0021 | 0.1604±0.0022 | 0.1491±0.0015 | 0.1625±0.0022 | **0.2447±0.0019** |

Table 2: AUC values on NYT 10 dataset. We run 10 models using random seeds with early stopping on the development set, and report the mean and standard deviation of test AUC values for all methods.

| AUC | NONE | ONE | ATT | AVE | ATT_RA | Ours |
|---|---|---|---|---|---|---|
| C=0.1 | 0.7316± 0.0069 | 0.7665±0.0087 | 0.7535± 0.0062 | 0.7499±0.0055 | 0.7514± 0.0053 | **0.7846±0.0066** |
| C=0.2 | 0.7246±0.0047 | 0.7610±0.0092 | 0.7472±0.0055 | 0.7428± 0.0052 | 0.7431±0.0071 | **0.7897±0.0059** |
| C=0.5 | 0.7251±0.0054 | 0.7605±0.0065 | 0.7453±0.0058 | 0.7409±0.0062 | 0.7423 ±0.0079 | **0.7915±0.0065** |
| C=1.0 | 0.7229± 0.0059 | 0.7559±0.0080 | 0.7424 ±0.0067 | 0.7368±0.0063 | 0.7395±0.0072 | **0.7942±0.0060** |

Table 3: AUC values on MIRGENE dataset. We run 10 models using random seeds with early stopping on the development set, and report the mean and standard deviation of test AUC values for all methods.

models using random seeds with early stopping on the development set. Table 2 and Table 3 show the mean and standard deviation test AUC values for each method on NYT 10 dataset and MIR-GENE dataset, respectively. We find that: (1) Our method significantly outperforms all baselines in federated settings. We believe the reason is that our denoising method can use cross-platform information to hinder false positive instances from poisoning local models, which leads to a better performance of the global model. (2) Directly applying FedAvg algorithm (McMahan et al., 2016) to the automatically labeled data achieve the worst results in both datasets. The reason behind that

is training on the noisy data will substantially hinder the performance of the model. Therefore, it is necessary to conduct denoise in federated distant supervision. (3) $C$ is the fraction of platforms that are activated on each round, which controls the amount of multi-platform parallelism. With increasing platform parallelism, the performance of all baselines declines slightly while our method performs better. Intuitively, increasing platform parallelism is able to lead to better results, since involving more platforms in training can increase the likelihood that all sentences with the same entity pair appear simultaneously. However, due to lack of cross-platform collaboration, all baselines handle label noise only based on its own local data, which may hamper the performance. In contrast, our method selects reliable instances among all activated platforms, which can effectively reap the benefits of increasing platform parallelism. (4) Leveraging attention mechanisms to denoise, an effective solution in centralized settings, seems not to work in federated settings. Compared with centralized training, the sentences in a bag scatter around different platforms in federated settings, so the number of the sentences with the same entity pair on a platform is small, which may lead to placing large attention weights on noisy sentences due to lack of inter-bag contrast.

## 4.5   INCREASING THE SIZE OF LOCAL DATASETS

In this section, we increase the size of local datasets by setting $K$ to 50. In such a way, each local dataset is twice as large as it was (when $K$ is set to 100). For a fair comparison, we fix $C = 0.1$, $B = 32$ and $E = 3$. Figure 4 show the results of AUC values. At the appendix, we also present corresponding precision-recall curves in Figure 7 and detailed precision values measured at different points along these curves in Table 10. From the results, we observe that: (1) Our proposed method significantly surpasses all baselines in both datasets. (2) Compared with setting $K$ to 100, the result of directly applying FedAvg algorithm (McMahan et al., 2016) to the automatically labeled data remains almost unchanged when $K$ is set to 50. (3) As the size of local datasets increases, all

| AUC | NYT | MIRGENE |
|---|---|---|
| NONE | 0.1325 | 0.7430 |
| ONE | 0.1856 | 0.7786 |
| ATT | 0.1806 | 0.7726 |
| AVE | 0.1687 | 0.7592 |
| ATT_RA | 0.1842 | 0.7639 |
| Ours | **0.2285** | **0.7941** |

Table 4: AUC values on NYT 10 dataset and MIRGENE dataset when $K = 50$.

denoising methods can achieve better results. The most likely reason is that compared with setting $K$ to 100, setting $K$ to 50 increases the probability that all sentences with the same entity pairs simultaneously exist in the same platform.

## 4.6   INCREASING THE NUMBER OF LOCAL UPDATES

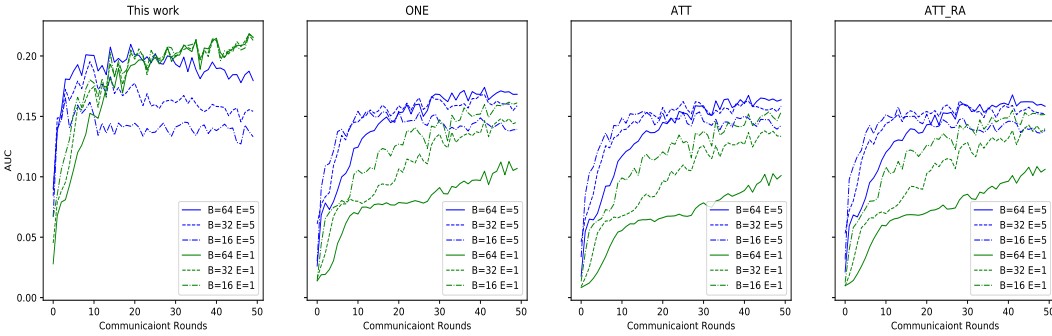

Figure 4: AUC values vs. communication rounds on NYT data with different $E$ (the number of local epochs) and $B$ (the local minibatch size).

In this section, we investigate the impact of varying the number of local updates in this section. The number of local updates is given by $E\frac{|D_i^*|}{B}$, where $|D_i^*|$ is the size of the denoised dataset in platform $i$ at a round, $B$ is the local minibatch size and $E$ is the number of local epochs. Increasing $B$, decreasing $E$, or both will reduce computation on each round. We fix $C$ to 0.1 and only $B$ and

$E$ are varied in this section [4]. The results are shown in Figure 4. We find that: (1) Compared with the other denoising baselines, our method converges faster to the optimal results. We conjecture that is due to that the proposed denoising method can effectively filter out the noise, which makes the relation extractor less affected by false positive instances and converge faster. (2) When setting $B$ to 64 and $E$ to 1, our method achieves the best AUC value. (3) Increasing the local minibatch $B$ may improve extraction performance. (4) Increasing the local epoch $E$ can speed up converge, but may not make the global model converge to a higher level of AUC value. These findings are in line with McMahan et al. (2016), which shows it may hurt performance when we over-optimize on the local dataset.

## 5 CONCLUSION

Considering data decentralization and privacy protection, we investigate distant supervision under the federated learning paradigm, which permits learning to be done while data stays in its local environment. To suppress label noise in federated settings, we propose a federated denoising framework, which can select reliable instances via cross-platform collaboration. Extensive experiments on two datasets have demonstrated the effectiveness of our method.

Distant supervision in federated settings is far from being solved and this work is just the beginning. There are still many problems need to be solved, such as noisy bag problem (Xu et al., 2013; Liu et al., 2017) and shifted label problem (Ye et al., 2019). Noisy bag problem means that all sentences containing the same entity pair are incorrectly labeled, and shifted label problem means the label distribution of training set does not align with that of test set. In federated settings, how these problems affect the relation extractor is still unknown. In our future work, we will devote to solve these problems in federated settings.

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

# A   CASE STUDIES

| Platform | Sentence | Type | NONE | ONE | ATT | ATT_RA | This Work |
|---|---|---|---|---|---|---|---|
| 10 | Most Muslims in Montenegro, mindful of the Serbs' killings of Muslims in Bosnia, are expected to vote to end ties with Serbia, but villagers in Podgorica are worried about how their Serb neighbors would react to separation. | Fasle Positive | ✓ | ✓ | ✓ | ✓ | ✗ |
| 7 | They have passed through Zagreb; Novi Sad; Belgrade; Pristina, in Kosovo; Skopje; Tirana, Albania; and Podgorica, Montenegro, on their way to Sarajevo. | False Positive | ✓ | ✓ | ✓ | ✓ | ✗ |
| 56 | This is a great day for the citizens of Montenegro to regain independence after 88 years, "said Ljubomir Djurkovic, a theater director from Centinje, a picturesque, pro-independence town to the west of Podgorica. | False Positive | ✓ | ✓ | ✓ | ✓ | ✗ |
| 26 | The time has come, " Montenegro's prime minister, Milo Djukanovic, said Thursday at a jubilant final rally in Podgorica, the capital. | True Positive | ✓ | ✓ | ✓ | ✓ | ✓ |

Table 5: A case to illustrate the effectiveness of the proposed model. A fact in KB is (*Podgorica*, /location/country/capital, *Montenegro*). Only the sentence in Platform 26 expresses the "/location/country/capital" relation, while the other sentences are all false positive.

Table 5 shows how different denoising methods select reliable instances in the training phase. In this case, a KB fact is (*Podgorica*, /location/country/capital, *Montenegro*). Aligning this KB fact with decentralized raw text generates four training instances, which are distributed in four different platforms. Only the sentence in Platform 26 correctly represents the "/location/country/capital" relation. The other sentences distributed in the other platforms are all false positive instances, which do not express the "/location/country/capital" relation.

From this case, we can find that: (1) If FedAvg algorithm (McMahan et al., 2016) was directly applied to the automatically labeled data, it would face a noisy environment where most sentences are false positive. (2) Previous denoising methods, such as ONE (Zeng et al., 2015), ATT (Lin et al., 2016) and ATT_RA (Ye & Ling, 2019), all fail to filter out false positive instances. In the worst cases, these methods will lose their denoising function. (3) Our proposed method can remove all false positive instances and only keep the true positive instance to train local models.

# B ADDITIONAL EXPERIMENTS

## B.1 CAN A STRONG EXTRACTOR MITIGATE THE LABEL NOISE IN FEDERATED SETTINGS?

In this section, we investigate the impact of involving a stronger extractor. More concretely, we replace the PCNN-based extractor with a BERT-based extractor (Devlin et al., 2018). In the BERT-based extractor, we use the architecture of entity mention pooling (Soares et al., 2019) to represent relations with the Transformer model (Vaswani et al., 2017), which is shown in Figure 5. Given a sentence $s$ and two entities within this sentence, we first segment the given sentence into tokens by the byte pair encoding (Sennrich et al., 2016) and feed these tokens into the BERT encoder. The output of the BERT encoder is the context-aware embeddings of tokens. After that, we use max pooling on the context-aware embeddings that correspond to the word pieces in each entity mention, to get two vectors $h_{e1}$ and $h_{e2}$ representing the two entity mentions. Finally, we concatenate these two vectors to get the representation of relation.

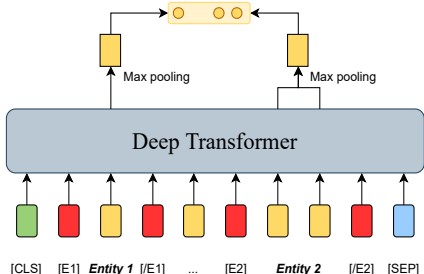

Figure 5: The main architecture for BERT-based extractor.

For a fair comparison, we fix $C = 0.1$, $B = 32$, $K = 100$ and $E = 3$. For the BERT-based extractor, we set the lr, lr decay and weight decay to 1e-5, 0.01 and 1e-5, and we use the pretrained BioBERT (Lee et al., 2019) and cased base version of BERT [5] as the initialization parameters in MIRGENE and NYT 10 dataset, respectively. The AUC values of PCNN-based extractor and BERT-based extractor on NYT 10 dataset and MIRGENE dataset are shown in Table 5. From the result, we find that: (1) Involving a stronger encoder can improve the performance for all denoising methods. (2) Whether leveraging PCNN or BERT as the encoder, our method significantly outperforms all baselines.

| Dataset | Extractor | NONE | ONE | ATT | AVE | ATT_RA | Ours |
|---------|-----------|------|-----|-----|-----|--------|------|
| NYT 10 | BERT-based Extractor | 0.1744 | 0.2217 | 0.2156 | 0.2120 | 0.2086 | **0.2526** |
| | PCNN-based Extractor | 0.1287 | 0.1719 | 0.1638 | 0.1521 | 0.1664 | **0.2189** |
| MIRGENE | BERT-based Extractor | 0.7510 | 0.7773 | 0.7798 | 0.7650 | 0.7768 | **0.8103** |
| | PCNN-based Extractor | 0.7316 | 0.7665 | 0.75335 | 0.7499 | 0.7514 | **0.7846** |

Table 6: The AUC values of PCNN-based extractor and BERT-based extractor on NYT 10 dataset and MIRGENE dataset when k is set to 100 and C is set to 0.1.

## B.2 ABLATION STUDY OF PARALLEL COMPUTATION

In this section, we conduct an ablation study of parallel computation in federated settings. Conventional federated learning approach (McMahan et al., 2016) is based on a master-slave topology [6]. In this topology, each platform (slave) trains a local model based on its own local data and a master server (master) coordinates massive platforms to collaboratively train a global model by aggregating these local model updates.

---

[5] https://github.com/google-research/bert
[6] https://en.wikipedia.org/wiki/Master/slave_(technology)

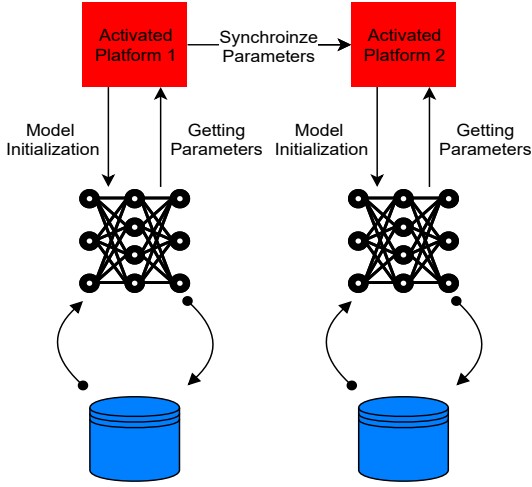

Figure 6: The main architecture for chain training.

To ablate parallel computation from the conventional federated learning topology, we propose a simple way, called chain learning, to handle decentralized data. In chain learning, we only train one local platform at a time, and then synchronize model parameters to the next platform for further training. We show the architecture of chain learning in Figure 6. Compared with federated learning, chain learning does not require a master server and parallel training.

|  | ONE | ATT | AVE | ATT_RA | Ours |
|---|---|---|---|---|---|
| Centralized Learning | 0.2323 | 0.2345 | 0.2147 | **0.2365** | - |
| Federated Learning | 0.1719 | 0.1638 | 0.1521 | 0.1664 | **0.2189** |
| Chain Learning | 0.1574 | 0.1538 | 0.1466 | **0.1590** | - |

Table 7: AUC values of different learning paradigms on NYT 10 dataset. In federated learning and chain learning, we fix $C = 0.1$, $K =100$ and $E =3$.

Table B.2 shows the AUC values of centralized learning, federated learning and chain learning on NYT 10 dataset. For a fair comparison, we set batch size $B$ to 32, set learning rate to 0.1, and set weight decay to 1e-5 for all learning paradigms. From the result, we find that: (1) Centralized learning achieves the best result compared to federated learning and chain learning; (2) Compared to federated learning, chain learning cannot achieve satisfactory results. We conjecture that catastrophic forgetting (McCloskey & Cohen, 1989; Goodfellow et al., 2013) may be the cause. In federated learning, averaging step for model integration is carried out in each epoch, which may ensures the generalization of the model. However, in chain learning, the model may forget the previous local data and overfit to the latest local data, which hampers the performance. (3) Our approach narrows the gap between federated training and centralized training in terms of performance. Therefore, we can conclude that not chain learning but cross-platform collaboration is the key to mitigate label noise in federated settings.

## C  BASELINES

---

**Algorithm 3** Federated Denoising Baseline

---

1: **Hyperparameters**: $K$ is the total number of platforms; $C$ is the fraction of platforms; $B$ is the
          local minibatch size; $E$ is the number local epochs; $\eta$ is the learning rate.
2: **Master server executes:**
3:    Initialize $\Theta_0$
4:    **for** communication round $q = 0,1,...$ **do**
5:       $m \leftarrow \max(C \times K, 1)$                              ▷ Select activated platforms
6:       $A_q \leftarrow$ (random set of $m$ platforms)
7:       **for** each platform $i \in A_q$ **in parallel do**
8:          $\Theta_{q+1}^k \leftarrow$ Local_Training$(i, \Theta_q)$
9:       $\Theta_{q+1} \leftarrow \sum_{i \in A_q} \frac{|D_i|}{\sum_{j \in A_q} |D_j|} \Theta_{q+1}^i$                ▷ Defined in Equation 5
10:
11: **Function** Local_Training$(i, \Theta)$:                       ▷ Run on platform $i$
12:    $\mathcal{B} \leftarrow$ (split $D_i$ into batches of size $B$)           ▷ A batch is a set of bag
13:    **for** each local epoch $e$ from 1 to $E$ **do**
14:       **for** batch $b \in \mathcal{B}$ **do**
15:          Conduct the denoising method       ▷ In **NONE**, we do not carry out this step
16:          Update $\Theta$ based on the gradients of the loss function
17:    **return** $\Theta$ to the master server

---

In Algorithm 3, we present the federated framework of denoising baseline. Compared with FedAvg algorithm (McMahan et al., 2016), we only add one step in local training to denoise. Compared with the proposed federated denoising framework, local platforms in the baseline framework handle label noise only based on its own local data.

# D  SOME TABLES AND FIGURES MENTIONED IN THE MAIN TEXT

| AUC | NYT | | | | | | MIRGENE | | | | | |
|---|---|---|---|---|---|---|---|---|---|---|---|---|
| | NONE | ONE | ATT | AVE | ATT_RA | Ours | NONE | ONE | ATT | AVE | ATT_RA | Ours |
| C=0.1 | 0.1318 | 0.1747 | 0.1658 | 0.1544 | 0.1695 | **0.2207** | 0.7436 | 0.7705 | 0.7696 | 0.7577 | 0.7597 | **0.7893** |
| C=0.2 | 0.1293 | 0.1747 | 0.1642 | 0.1531 | 0.1666 | **0.2315** | 0.7432 | 0.7649 | 0.7528 | 0.7491 | 0.7448 | **0.7923** |
| C=0.5 | 0.1305 | 0.1725 | 0.1657 | 0.1527 | 0.1647 | **0.2448** | 0.7365 | 0.7626 | 0.7516 | 0.7431 | 0.7484 | **0.7946** |
| C=1.0 | 0.1283 | 0.1715 | 0.1631 | 0.1503 | 0.1637 | **0.2465** | 0.7358 | 0.7570 | 0.7483 | 0.7432 | 0.7493 | **0.7966** |

Table 8: AUC values on NYT 10 dataset and MIRGENE dataset.

| P@N(%) | | NYT | | | | | | MIRGENE | | | | | |
|---|---|---|---|---|---|---|---|---|---|---|---|---|---|
| | | NONE | ONE | ATT | AVE | ATT_RA | Ours | NONE | ONE | ATT | AVE | ATT_RA | Ours |
| C=0.1 | p@100 | 57.0 | 63.0 | 60.0 | 57.0 | 62.0 | **69.0** | 83.0 | 87.0 | **89.0** | 87.0 | 86.0 | **89.0** |
| | P@200 | 49.0 | 60.0 | 57.0 | 55.0 | 55.5 | **67.0** | 75.0 | 79.5 | 77.5 | 78.0 | 77.0 | **80.0** |
| | P@300 | 44.7 | 54.7 | 52.7 | 53.0 | 53.3 | **63.0** | 69.0 | 71.3 | 69.3 | 70.7 | 71.3 | 70.7 |
| | Mean | 50.2 | 59.2 | 56.6 | 55.0 | 56.9 | **66.3** | 75.7 | 79.3 | 78.6 | 78.6 | 78.1 | **79.9** |
| C=0.2 | p@100 | 56.0 | 66.0 | 59.0 | 59.0 | 61.0 | **74.0** | 80.0 | 85.0 | 87.0 | 85.0 | 80.0 | **91** |
| | P@200 | 46.5 | 58.5 | 57.0 | 51.5 | 54.0 | **70.5** | 78.0 | 79.5 | 78.0 | 76.0 | 76.5 | **80.5** |
| | P@300 | 42.3 | 55.0 | 52.7 | 50.7 | 51.0 | **68.7** | 69.7 | 70.7 | 70.7 | 70.3 | 69.3 | **73.0** |
| | Mean | 48.3 | 59.8 | 56.2 | 53.7 | 55.3 | **71.1** | 75.9 | 78.4` | 78.6 | 77.1 | 75.3 | **81.5** |
| C=0.5 | p@100 | 47 | 65.0 | 63.0 | 58.0 | 60.0 | **77.0** | 79.0 | 87.0 | 87.0 | 84.0 | 83.0 | **92.0** |
| | P@200 | 47 | 59.0 | 57.5 | 53.5 | 54.5 | **74.5** | 75.5 | 80.0 | 75.0 | 75.0 | 77.0 | **82.5** |
| | P@300 | 44.3 | 55.0 | 53.3 | 52.7 | 50.3 | **71.7** | 70.3 | 70.7 | 70.0 | 70.3 | 71.0 | **74.0** |
| | Mean | 46.1 | 59.7 | 57.9 | 54.7 | 54.9 | **74.4** | 74.9 | 79.2 | 77.3 | 76.4 | 77.0 | **82.8** |
| C=1.0 | p@100 | 48.0 | 62.0 | 65.0 | 60.0 | 60.0 | **80.0** | 78.0 | 82.0 | 82.0 | 82.0 | 83.0 | **95.0** |
| | P@200 | 47.5 | 60.0 | 56.5 | 54.0 | 54.5 | **75.5** | 75.0 | 78.5 | 76.0 | 77.0 | 76.0 | **82.0** |
| | P@300 | 43.3 | 56.0 | 52.3 | 49.7 | 49.0 | **71.3** | 68.7 | 71.3 | 70.0 | 70.0 | 70.0 | **73.0** |
| | Mean | 46.3 | 59.3 | 57.9 | 54.6 | 54.5 | **75.6** | 73.9 | 77.3 | 76.0 | 76.3 | 76.3 | **83.3** |

Table 9: P@100, P@200, P@300 and the mean of them for each model in held-out evaluation on NYT 10 dataset and MIRGENE dataset.

| P@N(%) | NYT | | | | | | MIRGENE | | | | | |
|---|---|---|---|---|---|---|---|---|---|---|---|---|
| | NONE | ONE | ATT | AVE | ATT_RA | Ours | NONE | ONE | ATT | AVE | ATT_RA | Ours |
| P@100 | 53.0 | 63.0 | 65.0 | 63.0 | 69.0 | **73.0** | 82.0 | 90.0 | 88.0 | 84.0 | 85.0 | **94.0** |
| P@200 | 46.0 | 62.0 | 58.0 | 59.5 | 61.0 | **69.5** | 74.0 | 80.5 | 78.0 | 77.5 | 80.5 | **83.0** |
| P@300 | 45.0 | 59.3 | 54.7 | 56.7 | 59.0 | **68.7** | 69.7 | **71.7** | 70.7 | 70.7 | **71.7** | 71.0 |
| Mean | 48.0 | 61.4 | 59.2 | 59.7 | 63.0 | **70.4** | 75.2 | 80.7 | 78.9 | 77.4 | 78.6 | **82.7** |

Table 10: P@100, P@200, P@300 and the mean of them for each model in held-out evaluation on NYT 10 dataset and MIRGENE dataset when $K$ is set to 50 and $C$ is set to 0.1.

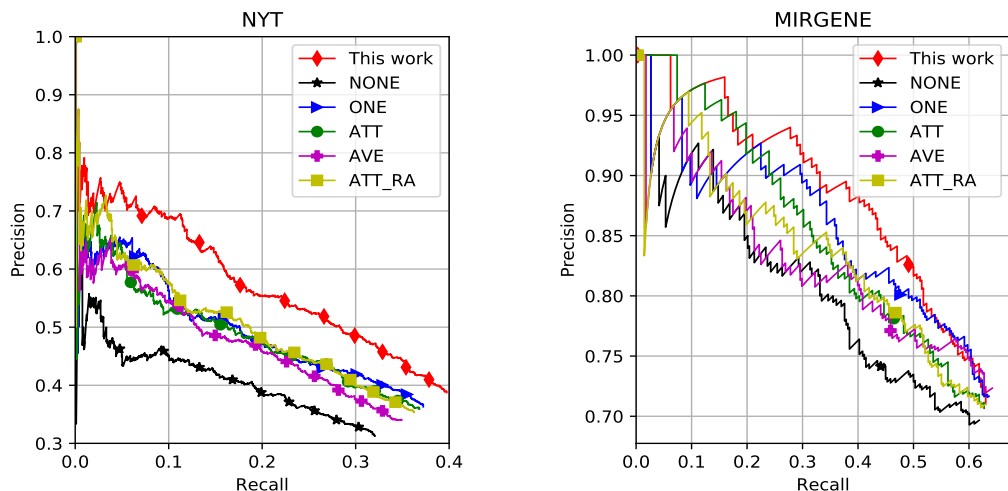

Figure 7: Aggregate precision-recall curves on NYT 10 dataset and MIRGENE dataset when $K$ is set to 50 and $C$ is set to 0.1.

