# OpenReview forum: "Distantly Supervised Relation Extraction in Federated Settings"
_ICLR.cc/2021/Conference — Reject_

### Official Review · AnonReviewer3 · 2020-10-27
**This paper tackles a new scenario for distant supervision relation extraction where extractions come from multiple private resources thus global training is not available.**

**Rating:** 5
**Confidence:** 4

**Review:**

Summary:
This paper introduced a new federated scenario for distantly supervised relation extraction where extractions come from multiple private resources. And therefore a global model cannot directly access all the data simultaneously.

To protect data privacy, this paper assigned each resource a local model for separate training. In order to apply the at-least-one sentence bag denoising technique, at the beginning of each round, local models will calculate scores of sentences in the local bag, compare scores with other resources and then generate the training data. After each round of parallel training, local models will be synchronized with the global model via a weighted average algorithm. By only sharing scores rather than the full text, data privacy is protected. Experiments over two datasets show promising results.

####################

Reasons for score:

I vote for marginally negative. Overall, this paper brought up an interesting challenge for this NLP sub-task. However, it is not clear to me the necessity of compromising accuracy while doing parallel computing, and model innovation is slightly weak.

####################

There are two comments from a performance-driven perspective:

1. Is there a specific reason for this task to do parallel computing? From my understanding (if correct), if parallel training is not required, we can avoid major information loss from delayed denoising and averaging step for model integration. In another word, if we only train one local platform at a time (and denoising frequently), and then synchronize model parameters to the next platform for further training, the performance should be close (or equal if using ONE model) to the performance upper bound when K=1.  If parallel computing is the key, then it is necessary to analyze the speed and performance trade-off.

2. It is necessary to use pre-trained language models like MTB (matching the blank) paper as the encoder because improvement over the current baseline could possibly be mitigated by a stronger encoder.

---

> ### Author Response · Authors · 2020-11-24
> **Replies to comments of AnonReviewer3**
>
> Thanks very much for your review and advice! They are really helpful.  Our replies to the questions are given as follows.
>
> [__Summary of the question__]:  Is there a specific reason for this task to do parallel computing?
>
> [__Our reply__]:  To answer this question，we design an ablation study of parallel computing, which is shown in Appendix B.  For convenience, the learning paradigm of training one local model at a time and then synchronizing the trained model parameters to the next platform for further training is called chain learning.  The results are shown in the following table.
>
> |                      |  ONE   |  ATT   |  AVE   | ATT_RA |  Ours  |
> | :------------------: | :----: | :----: | :----: | :----: | :----: |
> | Centralized Learning | 0.2323 | 0.2345 | 0.2147 | 0.2365 |   -    |
> |  Federated Learning  | 0.1719 | 0.1638 | 0.1521 | 0.1664 | 0.2189 |
> |    Chain Learning    | 0.1574 | 0.1538 | 0.1466 | 0.1590 |   -    |
>
> From the result,  we can conclude that parallel computing is not the key to mitigate label noise in federated settings.
>
> [__Summary of the question__]:  It is necessary to use pre-trained language models as the encoder.
>
> [__Our reply__]: We thank the reviewer for pointing out this issue.   In our revised version,  we provided the experiments of using a BERT-based extractor in Appendix B.  The results are shown in the following table.
>
> | Dataset |      Extractor       |  NONE  |  ONE   |   ATT   |  AVE   | ATT_RA |  Ours  |
> | ------- | :------------------: | :----: | :----: | :-----: | :----: | :----: | :----: |
> | NYT     | BERT-based Extractor | 0.1744 | 0.2217 | 0.2156  | 0.2120 | 0.2086 | 0.2526 |
> |         | PCNN-based Extractor | 0.1287 | 0.1719 | 0.1638  | 0.1521 | 0.1664 | 0.2189 |
> | MIRGENE | BERT-based Extractor | 0.7510 | 0.7773 | 0.7798  | 0.7650 | 0.7768 | 0.8103 |
> |         | PCNN-based Extractor | 0.7316 | 0.7665 | 0.75335 | 0.7499 | 0.7514 | 0.7846 |
>
> From the result, we find that: (1) Involving a stronger encoder can improve the performance for all denoising methods. (2) Whether leveraging PCNN or BERT as the encoder, our method significantly outperforms all baselines.

---

### Official Review · AnonReviewer1 · 2020-10-28
**An interesting paper but only contributes to a limited field.**

**Rating:** 5
**Confidence:** 4

**Review:**

This paper explores relation extraction with distant supervision in the federated setting and focuses on handling the label noise problem from automatic distant supervision by Multiple Instance Learning-based methods and proposes Lazy MIL.

For a specific triple (h, r, t) in KB, values (probability with relation r) for sentences containing h and t will be calculated locally in each platform, and such best value and index in each platform will be uploaded to the master server. A master server decides the most reliable sentence and broadcast such information to all activated platforms in this round.

Strengths:

+ This paper considers relation extraction in the federated setting, which is a new direction in this area;
+ The mentioned label noise problem is a real problem in distant supervision, and it’s intuitive that this problem exists in the federated setting;
+ Experiments show the effectiveness of the proposed method;

Weaknesses:

- The contribution of the paper is weak (in the context of the expectations of ICLR)
- For baselines in experiments, it’s not clear how to use baselines in the federated setting. For example, for “keep the other modules unchanged and only replace the denoising module” in the last sentence of Sec. 4.3, it’s confusing for me that how to replace this part. Is the part for broadcasting denoising information?
- Except for the experiment results, it’s not clear that if the proposed method is specific to the federated setting, and what are the federated-setting-specific designed compared with other baselines?

Questions:

In Sec. 4.4, “We believe the reason is that our denoising method can hinder false-positive instances from poisoning local models.” Does this mean that other baselines can’t hinder false-positive instances?

---

> ### Author Response · Authors · 2020-11-25
> **Replies to comments of AnonReviewer1**
>
> We thank the reviewer for the insightful comments and constructive feedback about the paper, and we would like to clarify several things to address the reviewer's concerns:
>
> [__Summary of question 1__]:  The contribution of the paper is weak
>
> [__Our reply__]:  In this paper, we investigate distant supervision under the federated learning paradigm, which is the first work to discuss this topic.  There are two reasons to conduct this research: (1) For distant supervision, previous studies are under the centralized training paradigm. However,  due to privacy restrictions, it is almost impossible or cost-prohibitive to conduct centralized training in practice.  (2) For federated learning,  prior work in federated learning has made the unrealistic assumption that the data stored by the local platforms are fully annotated with ground-truth labels [1, 2, 3], however,  the local datasets at the local platforms are often unlabeled, since annotating data requires both time and domain knowledge [4]. Distant supervision provides a compelling solution for automatic labeling a large scale dataset.
>
> In addition, since the automatic labeling in distant supervision inevitably accompanies with label noise, we present a very simple federated denoising method, which can select reliable instances via cross-platform collaboration. Through various experiments, the proposed method proved to be very effective in handling label noise in federated settings.
>
> [__Summary of question 2__]:    For baselines in experiments, it’s not clear how to use baselines in the federated setting.
>
> [__Our reply__]:  We apologize for the confusion. To describe baselines more clearly, we rewrote Section 4.3 and added the general algorithm for baselines in Appendix C.  We hope the algorithm can provide readers with a better understanding of baselines.
>
> [__Summary of question 3__]: It’s not clear that if the proposed method is specific to the federated setting, and what are the federated-setting-specific designed compared with other baselines?
>
> [__Our reply__]:  Compared with baselines,  we elaborately design a denoising method according to the characteristics of federated learning. In detail,  two characteristics of federated learning are used in our method. Firstly, local models and the global model share the same network architecture. Secondly,  the master server distributes the global model parameters to all activated platforms at the beginning of each round, and the activated platforms use these parameters to initialize the local models. At this moment, all activated local models share the same model parameters.  In lazy one algorithm, we make full use of this moment. More concretely, with the same model parameters and the same network architecture, the outputs of different local models can be compared with each other. Therefore, we ask each activated platform to upload the value and index of the instance with the maximum conditional probability associated with a given triple. After that, the master server decides which local instance can be selected among all activated platforms based on the uploaded value.
>
>
>
> [__Summary of question 4__]:  “We believe the reason is that our denoising method can hinder false-positive instances from poisoning local models.” Does this mean that other baselines can’t hinder false-positive instances?
>
> [__Our reply__]:  We apologize for the confusion. In our revised version, we have modified this flawed statement. Among baselines, NONE cannot hinder false-positive instances, the others can only hinder false-positive instances according to the local data of a single platform.  Compared with these baselines, our method can coordinate all activated platforms to jointly remove false-positive instances.
>
> [1] Towards utilizing unlabeled data in federated learning: A survey and prospective
>
> [2] Benchmarking semi-supervised federated learning
>
> [3] Federated self-supervised learning of multi-sensor representations for embedded intelligence
>
> [4] Exploiting unlabeled data in smart cities using federated learning

---

### Official Review · AnonReviewer2 · 2020-10-29
**Interesting work, Lacks novelty and unclear experimental setup**

**Rating:** 6
**Confidence:** 4

**Review:**

Summary:

This work focuses on investigating distant supervision for relation extraction task within federated learning paradigm. The proposed Lazy Multi-instance Learning approach identifies reliable sentences for the same entity pairs across platforms to denoise distantly supervised data and perform relation extraction.

The proposed approach has been applied on two relation extraction datasets and has reported gains.


Strengths:
- The core of distant supervision for relation extraction is to denoise sentences for building a supervised classifier. The problem is well investigated in NLP community. However, this work focuses on modeling the distant supervision problem in real-world environment addressing data decentralization, ownership and privacy. I buy the idea of applying the existing methods or extend the NLP applications in federated setup.
- the paper is well written, clearly motivation and addressed real-world problem
- interesting work: blend of distant supervision and federated learning for NLP task

Weaknesses:
- the paper lacks novelty in terms of methodology
- Distant supervision for relation extraction (PCNN and MIL) and the federated learning methodologies used in this work have been inspired from existing works. This works combines the two.
- the experimental setup needs more clarity


Questions:
1. Assumption made: Sentences with same entity pairs must scatter across platforms. How would the Lazy MIL system bootstrap without such an assumption (no same entity pairs)?
2. Assumption made: The number of output classes are same therefore the \Theta parameter (including output-layer weights connecting softmax) is used across platforms. In real world, it is not the case. Each platform (or customer) may have different relation types. How would you aggregate/distribute the output layer parameters?
3. How to obtain a minimum viable global \Theta? How to init this without having the actual training data in Master server?
4. In equation 2, the v^i is not controlled by any threshold. Will it denoise if its value is too low (say <0.50) and it is still the highest among all the platforms?
5. In section 4.1 in data partitioning, how do you ensure the number of relation types remains same across all platforms?

Additional comments:

- Include an ablation study analyzing scores due to different values of K.

Experimental setup  unclear:
- how do to init \Theta in global model?
- What is the held-out data set for local and global model training?

Results:
- the experimental results show noticeable gains. It is surprising as the NYT dataset is well investigated in distant supervision settings.
- As the training data is split across platforms, the overall system is decoupled into several local models. In essence, the overall performance in federated settings should deteriorate or remain competitive to the baseline models due to 'no joint' training on the overall corpora as well as due to (somewhat) lossy aggregation. Please provide a detailed reasoning about the noticeable gains achieved.

Reproducibility
- no code available

---

> ### Author Response · Authors · 2020-11-25
> **Replies to comments of AnonReviewer2**
>
> We thank the reviewer for the insightful comments and constructive feedback about the paper, and we would like to clarify several things to address the reviewer's concerns. To make these replies more clear, we first answer question 2, then question 3, next question 4, then question 1 and finally question 5.
>
>
> [__Summary of question 2__]:  Each platform may have different relation types.
>
> [__Our reply__]:  This is a very realistic question. (1) In conventional federated learning [1], researchers usually assume that the total number of labels among all platforms is known. We think this assumption is reasonable in most cases. If we can know the total number of labels, we can set the output dimension as the total number of labels, so that we can solve the problem that each platform has different relation types. (2) If the total number of labels is unknown, we think this will be a  harder setting.  But there are some studies devoted to this topic, such as [2, 3, 4].  In this paper, we follow the assumption that the total number of labels is known. In our future work, we will explore the setting that the total number of labels is unknown.
>
> [__Summary of question 3__]:  How to initialize the global model without having the actual training data in the master server?
>
> [__Our reply__]:  We apologize for the confusion. In the PCNN-based extractor, we first use the pretrained word embeddings to initialize the embedding module. After that, we randomly initialize the remaining modules. In the BERT-based extractor, we use the pretrained BERT model weight to initialize the encoder, and randomly initialize the output-layer weight.  In initialization, there is no need to access data.
>
> [__Summary of question 4__]:  In equation 2, the $v^i$ is not controlled by any threshold. Will it denoise if its value is too low (say <0.50)
>
> [__Our reply__]:  This is a very insightful question. In this paper, we follow the *at least one assumption* [5, 6]: if two entities preserve a relation in a KB, at least one sentence that mentions the entity pair expresses the relation.  In this assumption, even if the $v^i$ of an instance is low and it is still the highest among all the platforms, we still keep this instance as a true-positive instead of removing it. This assumption is not perfect, because there is a case that all sentences containing the same entity pair are incorrectly labeled, which is called the noisy bag problem [7, 8]. In this paper, we ignore the noisy bag problem,  like most previous studies, since label noise is the leading problem in distant supervision.
>
>
> [__Summary of question 1__]:  Sentences with the same entity pairs must scatter across platforms. How would the Lazy MIL system bootstrap without such an assumption (no same entity pairs)?
>
> [__Our reply__]:  This is an interesting question. We think all baselines and Lazy MIL will degenerate into NONE in this case. More concretely, sentences share no same entity pairs, which means that each bag only contains one sentence. In such a case, all noise reduction methods will not function, if we follow the *at least one assumption*. In this case, the leading problem in distant supervision change from label noise to noisy bag.
>
> [__Summary of question 5__]:   how to ensure the number of relation types remains the same across all platforms
>
> [__Our reply__]:  Following previous work [1], the training data are shuffled and then partitioned into K platforms in this paper. We do not ensure the number of relation types remains the same across all platforms, but need to know the total number of labels among all platforms.
>
> [__Summary of question 6__]:   Reproducibility, no code available
>
> [__Our reply__]:  The code will be publicly available soon.
>
> [1] Communication-efficient learning of deep networks from decentralized data
>
> [2] Personalized Federated Learning: A Meta-Learning Approach
>
> [3] FedNER: Privacy-preserving Medical Named Entity Recognition with Federated Learning
>
> [4] Improving Federated Learning Personalization via Model Agnostic Meta Learning
>
> [5] Modeling relations and their mentions without labeled text
>
> [6] Knowledge-based weak supervision for information extraction of overlapping relations
>
> [7] Filling knowledge base gaps for distant supervision of relation extraction.
>
> [8] A soft-label method for noise-tolerant distantly supervised relation extraction.

---

### Official Review · AnonReviewer4 · 2020-10-30
**A good problem but improvements are expected**

**Rating:** 4
**Confidence:** 4

**Review:**

This paper addresses relation extraction problem for distributed platforms for privacy concerns. The authors propose to leverage federated learning with denoising techinques for better performance. The relation extractor is set to the conventional piece-wise CNN and the main contributions focus on dealing noises in the distributed settings.

Pros:
* A new method for relation extraction in the federated learning to address privacy concerns.
* Best performance compared to baseline models.

Cons:
* Lack of novelty. This method follows the federated learning paradigm with few improvements. The main contribution is limited to the proposed lazy multiple instance learning, which is not specific to relation extraction problem. If this is not specified for relation extraction, more tasks are expected to demonstrate it is a general algorithm.
* Experiments are insufficient. First, it is doubtful whether the i.i.d. setting is a practical assumption for relation extraction, because in the real world the quality of corpus in different platforms may vary drastically. Second, given the i.i.d. setting, the authors can repeat their experiments to reduce randomness, because the proposed denoising strategy may only perform well in situations that useful sentences are allocated to only few platforms. The authors may provide more analysis on the distribution of contributions from platforms.
* Writing could be improved. In section 3.3, what does the ``value v^i'' refers to? Also, if v^i > v^j, then the id^i-th sentence in the platform i is selected...''. Does it mean that at every round only one sentence from a platform is selected? What if for a certain relation, there are many promising candidate sentences from one platform?

---

> ### Author Response · Authors · 2020-11-25
> **Replies to comments of AnonReviewer4**
>
> We thank the reviewer for the insightful comments and constructive feedback about the paper, and we would like to clarify several things to address the reviewer's concerns:
>
> [__Summary of question 1__]:   Lack of novelty. The main contribution is limited to the proposed lazy multiple instance learning
>
> [__Our reply__]:  In this paper,  we investigate distant supervision under the federated learning paradigm. To our best knowledge, combining federated learning with distant supervision is still an unexplored territory. This work can benefit to both distant supervision and federated learning. For distant supervision, our work extends distant supervision to federated settings, which can address data decentralization, ownership and privacy in a real-world environment. For federated learning, we release the unrealistic assumption that the data stored by the local platforms are fully annotated with ground-truth labels  [1, 2, 3]. In this work, the local datasets can be unlabeled and we use a knowledge base to automatically label local datasets.
>
> [__Summary of question 2__]:   it is doubtful whether the i.i.d. setting is a practical assumption for relation extraction, because in the real world the quality of corpus in different platforms may vary drastically
>
> [__Our reply__]:  I agree with the reviewer.  In our preliminary thinking,  we have also considered involving the non-IID case.  There are two reasons to only present the results of IID setting. (1)  It is difficult to design an appropriate non-IID sampling strategy for federated distant supervision.   if we follow previous studies [4, 5] to sample non-IID data,  the federated distant supervision will degenerate into a simpler setting. In this setting,  sentences in a bag will have a high likelihood to exist in the same platform.  (2) The IID setting can perfectly reveal the key challenge in federated distant supervision, which is the sentences in a bag scatter around different platforms and we need to use cross-platform information to denoise.
>
>
>
> [__Summary of question 3__]:   Given the i.i.d. setting, the authors can repeat their experiments to reduce randomness
>
> [__Our reply__]:  We thank the reviewer for pointing out this issue. In our revised version, we repeated each method 10 times in the main experiments and reported the mean and standard deviation of test AUC values for these methods. The results are shown in Table 2 and Table 3.
>
> [__Summary of question 4__]:  Does only one sentence from a platform is selected at every round?
>
> [__Our reply__]:  We apologize for the confusion.  At each round, for a given triple $(h,r, t)$ in KB, only one sentence that contains the entity pair $(h, t)$ is selected among all activated platform. Assume that there are $N$ entity pairs in KB ($N$ is 96, 678 in the NYT training set),  $N$ sentences are selected among all activated platforms in each round. Therefore,  for a certain platform, the number of selected sentences is a variable, ranging from 0 to $N$.
>
> [__Summary of question 5__]:  What if for a certain relation, there are many promising candidate sentences from one platform?
>
> [__Our reply__]:   This is an interesting and insightful question.  In this paper, we only select the most likely sentence among all the activated platforms for each entity pair,  therefore,  the proposed method may lose information containing in neglected sentences.  However, through various experiments, we find that ONE and our method perform better than the other baselines. We conjecture that involving too many uncertain instances in training may hamper the performance in federated settings.
>
> [1] Towards utilizing unlabeled data in federated learning: A survey and prospective
>
> [2] Benchmarking semi-supervised federated learning
>
> [3] Federated self-supervised learning of multi-sensor representations for embedded intelligence
>
> [4] Communication-Efficient Learning of Deep Networks from Decentralized Data
>
> [5] Federated Learning with Non-IID Data

---

### Official Review · AnonReviewer5 · 2020-11-05
**Simple and probably efficient idea, some questions about evaluation**

**Rating:** 6
**Confidence:** 3

**Review:**

Summary

The paper investigates intersection of federated learning and distant supervision of knowledge graphs from texts. The main innovation is a simple yet empirically effective denoising rule that selects only the sentences deemed to be the most reliable for learning in each round of training.


Strong points
* The method is very simple to implement therefore it might be widely adopted as a baseline.
* The denoising step is specific only for federated learning and distant supervision. It can be applied to many other domains as well (and not just for relation extraction).

Weak points
* The paper would benefit from further proofreading.
* Some uncertainty about experimental results, namely repeatability of the runs and methodology of hyperparam selection (more in questions section).


Recommendation

* Pre rebuttal: I like the simplistic approach however in current form I am leaning towards rejecting the paper due to my uncertainty in the empirical evaluation. However I am willing to change my evaluation if these questions are resolved.

* Post rebuttal update:  Several of my concerns about clarity of the experimental section were addressed. Therefore I increased my score and now I am inclining towards accepting the paper.

Questions

* Methodology of obtaining the empirical results isn't clear from the text. Section 4.1 mentions a single held out test set. However then it isn't clear how hyperparam sweep discussed in Sec 4.2 was done. If there is only a single test set were the hyperparams selected on the same test set that is later used to report results in Fig 2 and 3? I would consider that a flaw. Or is there a second validation set (that isn't mentioned in the text explicitly) used to tune hyperparams?

* Curves in Fig 2 and 3 seem to be from just one run for each setting. Doing more runs with different random initializations and reporting mean (or top k results) + confidence intervals would increase confidence in these results.


Possible improvements

* "Lazy MIL almost does not leak the corpus information in each platform" --- can you make this statement more precise?

* Baseline attention models are always forced to spread their attention among sentences in the local bag even if they are all irrelevant. What if the attention module is allowed to output "all sentences are irrelevant" option (this is sometimes called 'sentinel', https://arxiv.org/abs/1612.01887). This might work as learnable denoising that deals with the issue that some local bags don't include any relevant examples.

* Add references for all datasets in section 4.1.
* Alg 1: are lines 6 and 7 needed? they seem to be contained by line 8.
* Alg 1, line 11: this piece of Python in pseudocode might not be readable by everyone, I would explain it in plain words.
* Alg 2: Hyperparams --- "E is the number" (what number?)

Some language issues:

* leads to catastrophic repro- ducibility -> is not reproducible
* is a large biomedical with -> biomedical dataset with?
* KB is public available -> publicly available
* the token representation is represented as -> the token is represented as?
* The overall of Lazy MIL is illustrated in Algorithm 1. --- overall design of? overview of?
* "to extract the high-level sentence representation from three segments of CNN outputs, and the boundaries of segments are determined by the positions of the two entities." -> I would first define what segments are and only after that how they are used in NN, here the order is reversed and I find it more difficult to follow.
* Konečný (different accentation in the last character)

---

> ### Author Response · Authors · 2020-11-24
> **Replies to comments of AnonReviewer5**
>
> Many thanks to the reviewer for giving these insightful comments about the paper.
> Our replies to the questions are given as follows.
>
> [__Summary of question 1__]:  Methodology of obtaining the empirical results isn't clear from the text. Section 4.1 only mentions a single held out test set. However, then it isn't clear how hyperparam sweep discussed in Sec 4.2 was done.
>
> [__Our reply__]:  We apologize for the confusion. Actually, there is a  second validation set.  Take NYT 10 as an example. The development set contains 55167 sentences, 28077 entity pairs and 1,808 relational facts.  In our experiments, the optimal hyperparameters are determined by grid search based on the performance of the development set. After determining the optimal hyperparameters, we use the selected hyperparameters to report the results on the test set. In our revised version, we rewrote Section 4.1 to explicitly provide the information of the development set.
>
>
> [__Summary of question 2__]:  Doing more runs with different random initializations and reporting mean + confidence intervals would increase confidence in these results.
>
> [__Our reply__]:  We thank the reviewer for pointing out this issue.  In our revised version, we re-conducted the main experiments with different random seeds and reported the mean and standard deviation of test AUC values for each method. The results are shown in Table 2 and Table 3.
>
>
> [__Summary of question 3__]:  Lazy MIL almost does not leak the corpus information in each platform
>
> [__Our reply__]:   Although the training examples never leave the local platform，the server can infer the distribution of entities in each local platform from the uploaded information. In detail, for a given triple $(h, r, t)$, if a local platform uploads the value $v$ and index $id$ to the master server, the master server will infer that the entity $h$ and the entity $t$ exist on that platform.
>
>
> [__Summary of question 4__]:  Baseline attention models are forced to spread their attention among sentences in the local bag even if they are all irrelevant. What if the attention module is allowed to output "all sentences are irrelevant" option?
>
> [__Our reply__]:  This seems to be a very interesting and promising idea to improve the attention mechanism in federated denoising. We have tried to implement this approach，but found it is not easy to assign appropriate supervised signals to these sentinels. From my understanding (if correct), if the model is allowed to output "all sentences are irrelevant" option in distant supervision,  there must be "all sentences are irrelevant" supervised signal in the training set. We think combining sentinel with reinforcement learning may be a solution. Since this work is just the beginning of federated distant supervision, we will continue to study how to improve the attention mechanism in federated denoising.
>
>
> [__Summary of question 5__]:  Language issues:  Language issues
>
> [__Our reply__]:  We thank the reviewer for pointing out these issues.  In our revised version, we polished the article again to avoid such language issues.

---

### Author Response · Authors · 2020-11-24
**General reponse to all comments**

Thanks very much for the comments from all the reviewers. In the updated version, we mainly update the paper in the following aspects:

1. To reduce randomness，we re-conducted the main experiments and reported the mean and standard deviation of test AUC values for each method. The result is shown in Table 2 and Table 3.
2. Added Appendix B to provide the experiments of using a BERT-based extractor, as well as an experiment to ablate parallel computing.
3. Added Appendix C to provide the algorithm of baselines
4. Rewrote Section 3.2 to describe the relation extractor more clearly.
5. Rewrote Section 4.1 to provide the information of the development set
6. Polished the article to avoid language issues

We hope the discussions added to those appendices can provide readers with a better understanding of our method.

---

### Decision · Program_Chairs · 2021-01-07
**Final Decision**

**Decision:**

Reject

**Comment:**

The paper studies Distantly Supervised Relation Extraction(DSRE) in distributed settings. Though DSRE has been studied in Centralized setting it has not been studied in distributed platform. This  paper leverages the federated learning setup for this problem and proposes to use Lazy MIL for this purpose.  The paper  identifies the main challenge as label noise but does not attempt to characterise the severity of the problem vis-a-vis the centralised setup.  Though intuitive but a formal approach would have helped in understanding the importance of the derived results better.